# Hierarchical Self-Assembly and Conformation of Tb Double-Decker Molecular Magnets: Experiment and Molecular Dynamics

**DOI:** 10.3390/nano13152232

**Published:** 2023-08-01

**Authors:** Patrick Lawes, Mauro Boero, Rabei Barhoumi, Svetlana Klyatskaya, Mario Ruben, Jean-Pierre Bucher

**Affiliations:** 1Institut de Physique et de Chimie de Matériaux (IPCMS), Université de Strasbourg, UMR 7504, F-67034 Strasbourg, France; patrick.lawes@ipcms.unistra.fr (P.L.); mauro.boero@ipcms.unistra.fr (M.B.); rabiybarhoumi@hotmail.fr (R.B.); 2Institute of Nanotechnology and Institute of Quantum Materials and Technology (IQMT), Karlsruhe Institute of Technology, 76131 Karlsruhe, Germany; svetlana.klyatskaya@kit.edu (S.K.); mario.ruben@kit.edu (M.R.); 3Centre Européen de Science Quantique (CESQ), Institut de Science et d’Ingénierie Supramoléculaires (ISIS), Université de Strasbourg, F-67083 Strasbourg, France

**Keywords:** 2D self-assembly, molecular magnets, bis(phthalocyaninato) terbium (III), STM, electronic contrast, ab initio, molecular dynamics, molecular conformation, double-decker, quantum computation

## Abstract

Nanostructures, fabricated by locating molecular building blocks in well-defined positions, for example, on a lattice, are ideal platforms for studying atomic-scale quantum effects. In this context, STM data obtained from self-assembled Bis(phthalocyaninato) Terbium (III) (TbPc_2_) single-molecule magnets on various substrates have raised questions about the conformation of the TbPc_2_ molecules within the lattice. In order to address this issue, molecular dynamics simulations were carried out on a 2D assembly of TbPc_2_ molecules. The calculations are in excellent agreement with the experiment, and thus improve our understanding of the self-assembly process. In particular, the calculated electron density of the molecular assembly compares well with STM contrast of self-assembled TbPc_2_ on Au(111), simultaneously providing the conformation of the two Pc ligands of the individual double-decker molecule. This approach proves valuable in the identification of the STM contrast of LnPc_2_ layers and could be used in similar cases where it is difficult to interpret the STM images of an assembly of molecular complexes.

## 1. Introduction

Single-molecule magnets (SMMs) have been the object of intense research due to their potential application in magnetic data storage, molecular spintronics, and as qubits for quantum computation [1]. However, in practice, addressing individual SMMs, for example, by means of the tip of an STM, requires the precise control of the 2D organization of these molecules on a surface [2]. This question is of great relevance since intermolecular interactions and molecule–substrate interactions play important roles in defining the properties of assembled monolayers. Part of these properties can be tuned by functionalizing the molecular building blocks with proper functional groups in order to drive molecular conformation and impact the spin properties of the SMMs. In this context, the lanthanide double-decker complexes were found to exhibit interesting SMM behaviors at low temperatures [3,4,5,6,7,8,9,10] without exhibiting the disadvantages of other systems [11]. Double-decker complexes involving the phthalocyanine sandwich approach have been extensively investigated. As a matter of fact, the significant breakthroughs recorded in the field of molecular magnets have been lanthanide-based, which can mostly be attributed to the large anisotropy arising from the unquenched orbital angular momentum of the f−orbitals.

Due to their spin properties, the lanthanide double- and triple-decker molecules gained most relevance in the realization of qubits and qudits for use in quantum information processing (QIP) [12,13,14]. Developments accelerated when a reliable electric readout of the quantum information of single-spin qubits was proposed. In this context, the bis(phthalocyaninato Tb(III) (TbPc_2_) single-molecule magnet is one of the well-known examples in which the delocalized π-radical electron spin of the Pc ligand allows the reading of the electronic and nuclear spin states of the Tb qubits [15,16]. It has been evidenced that the π-radicals also play a major role in the quantum mechanical associations of such SMMs [6].

Structurally, TbPc_2_ features a central Tb^3+^ octa-coordinated ion sandwiched between two parallel phthalocyanines, with a dihedral rotation angle between the Pc ligands close to ϑ = 45°. This leads to a square-antiprismatic (D_4d_) coordination geometry [3,4] (see Figure 1a,b). Whereas the dihedral angle of 45° is adopted by the stable gas-phase molecules, it is also the usual conformation for isolated molecules on surfaces [5,6,17]. However, it was inferred from a series of STM measurements conducted on monolayers of lanthanide double-deckers (LnPc_2_) on surfaces that the molecules can also adopt a ϑ = 30° dihedral angle (or some other values different from 45°) [5,6,7,8], leading, for example, to a square checkerboard arrangement of 45° and 30° molecules.

The reasoning was as follows. It was assumed that the lower Pc ligands, those in contact with the substrate, all adopt the same orientation, one similar to the those observed in single metallo-phthalocyanine monolayers [18]. Since the upper Pcs of the double-decker adopt different orientations inside the molecular layer (STM observation), it was then necessary to postulate the existence of at least two conformations with different dihedral angles for the molecules constituting the layer. However, these conclusions came only as a best guess and were not based on any direct observations since only the top Pc ligand is “visible” in STM experiments (Figure 1b). As a result, this assumption has never been verified properly. It is noteworthy that other works on monolayers of LnPc_2_ did not claim such an alternation in the dihedral angle [6,8,9,10]. In particular, it has been found that 2D islands, made up of a small number of TbPc_2_ molecules (one to four) on Au(111), follow a well-defined arrangement with ϑ = 45° that optimizes compactness [6]. It is the purpose of this work to find out the extent to which the association of molecules in networks can modify their conformation and hence their electronic and magnetic properties.

## 2. Materials and Methods

### 2.1. Scanning Tunneling Microscopy and Spectroscopy

All sample preparations were carried out in an ultrahigh vacuum (UHV) system with a base pressure of 1 × 10^−10^ mbar. The single crystalline Au(111) substrate was cleaned by Ne^+^ sputtering and annealing cycles. The powder sample of TbPc_2_ molecules (synthesized by the Ruben group at the Karlsruhe Institute of Technology) was first degassed in vacuo for several hours at a temperature slightly below the sublimation temperature. Deposition of TbPc_2_ occurred at a sublimation temperature of 600 K onto the substrates kept at room temperature. Molecule sublimation was performed in a side chamber of the UHV system; during this operation, the pressure was kept below 1 × 10^−9^ mbar. All STM/STS data were acquired at 4.5 K. STS spectra were measured using lock-in detection, with a modulation between 1 mV and 10 mV (rms) depending on the features to be resolved. Manipulation by means of the STM tip of this category of molecules was described earlier [17,18,19].

### 2.2. Computational Method

All calculations have been conducted in the framework of the density functional theory [20] (DFT) as implemented in the CPMD [21] code. The exchange and correlation functionals adopted are the ones of Becke [22] and Lee-Yang-Parr [23], respectively. These are complemented by exact exchange [24], as routinely performed in the hybrid functional B3LYP. For the long-range van der Waals interactions, we resorted to Grimme’s D2 formula [25]. This van der Waals correction was preferred to that of D3, which proved to be responsible for parasitic effects that even lead to a non-physical phase separation in condensed matter systems [26]. In fact, except for D3, a similarity in performance is found for D2, rVV10 and several others [27]. Core–valence interactions were described by norm-conserving Troullier–Martins [28] pseudopotentials (PPs) for N, C, and H, while for Tb we make use of a Goedecker–Teter–Hutter [29] semicore PP. Valence electron orbitals were expanded on a plane wave (PW) basis set, with a cut-off energy of 80 Ry. A spin-unrestricted approach was adopted, and the structure was fully optimized via damped dynamics [30], as implemented in the developer’s version of the CPMD code, until residual atomic forces were smaller than 10^−4^ Hartree/Bohr. The annealing factor of the ions was set to 0.995, and we used an integration time step of 4.0 (0.097 fs) and a fictitious electronic mass of 340 au for the propagation of the electronic wavefunctions within a Car–Parrinello scheme. A total simulation time of about 8 ps was needed to bring the system to a stress-free structure at 0 K.

## 3. Results and Discussion

In a previous work, it was found that interactions between molecules in a cluster of less than 10 molecules are such that only two types of molecular arrangements are found to correspond to the parallel (dimer in Figure 2a) and staggered (see the trimer in Figure 2b) arrangements. In either case, it has been unambiguously determined that each TbPc_2_ molecule conserves its ϑ = 45° dihedral angle (for details, see Ref. [6]). Additionally, the quantum mechanical behavior of molecular π-radicals in the cluster formation has been emphasized and detected by the Kondo resonance measurement with atomic resolution [6]. With increasing island size, however, molecules have to accommodate multiple interactions and therefore adopt a more complex geometrical arrangement. This was observed using STM on a semi-infinite 2D domain (lower left of Figure 2a–c), the understanding of which is the main purpose of the present work. A simple scenario could be that the relatively strong molecule–molecule interaction in clusters of a few molecules is progressively destabilized when the 2D lattice is formed due to interaction with the ensemble of surrounding molecules.

The square checkerboard lattice mentioned above for the monolayers of TbPc_2_ molecules on Au(111) manifests itself by alternating bright and dark STM contrasts located in the center part of the molecules, as shown in Figure 2 [5,6,7]. Here, we simply term them A and B, respectively (Figure 2c). It was observed, however, that these contrasts only appear at specific values of the bias voltage [7], indicating a possible spectroscopic origin. For example, the STM image of Figure 2c has been acquired at −0.5 V and shows contrasts that are in good agreement with those observed by others on the same system and in the same bias voltage interval [5]. In addition, the HOMO–LUMO gap measured at 760 meV for a single TbPc_2_ molecule on Au(111) [6] is strongly decreased for spectra acquired above the 2D molecular lattice (Figure 2d). The HOMO–LUMO gap is only 100 and 200 meV (dotted vertical lines) for A and B molecules, respectively. The reduction in the gap occurs due to the molecular orbital overlap induced by the 2D lattice formation. However, at this stage, and in the absence of specific hints, it remains impossible to identify unequivocally the origin of the STM contrast. One should add that, to date, no calculation has been able to fully explain the origin of A and B contrasts observed in STM experiments. The difficulty in doing so comes from the large number of atoms to be considered in the unit cell of a semi-infinite system. In this work, we address this issue by means of a full molecular dynamics and ab initio DFT calculation, taking into account the symmetry of the system and the weak interaction observed between the TbPc_2_ molecules and the Au(111) substrate.

For the purpose of structure optimization, we initiate the calculation with an assembly of four TbPc_2_ molecules with dihedral angles of 45° (Figure 3a). The only experimental input is the dimension of the square lattice (1.42 nm). Periodic boundary conditions are adopted for the calculation. For the optimization of the molecular dynamics process, we assert that the molecule–substrate interaction can be neglected in the first approximation based on the mobility of compact clusters, such as tetramers on Au(111), which are tested by means of STM manipulation [6]. This approach is validated by the stability of the molecular radical on Au(111), as probed via Kondo physics (absence of charge transfer from the substrate). It should be stated, however, that for other substrates (different systems), the molecule–substrate interaction is not negligible in the calculations [7,9,10,31]. After the full annealing of the system, it is found that the tetramer keeps its quasi-flat 2D configuration (Appendix A) and, above all, that the dihedral angles of all molecules remain unchanged and equal to 45° (Figure 3a). In addition, it is found that the molecules adopt two different azimuthal orientations for A and B. The A and B orientations are found along either diagonal of the square lattice (see Figure 3a). Simulation was initiated from other configurations, but always converged to the one shown in Figure 3a, with 45° dihedral angles for both A and B sites.

For comparison with the STM experiment, Figure 3b shows the calculated electron density within an interval [*E*_F_−0.5 eV, *E*_F_]. It can be noted that the external lobes correspond well with the STM image of Figure 2c. The agreement with the experiment also holds for the molecular orbital overlap between two adjacent molecules, as highlighted by the circle in the electron density image of Figure 3b. In addition to the eight external lobes of the upper Pc, the electron density clearly shows that eight small inner lobes merge into a cross-like contrast (Figure 3b). These results compare well with the experiment for data taken within the same energy range as that of the applied sample bias [6]. The results reflect the checkerboard symmetry of the system in that A-type molecules exhibit one azimuthal orientation, whereas B-type molecules exhibit another azimuthal orientation. From the contrast of the inner lobes, it is easily found that the whole B molecule is rotated by 30° clockwise with respect to the A molecule. These results are in perfect agreement with respect to the A molecule, in good agreement with the experiment [6].

It is instructive to consider the images obtained for different values of electron density. It is found that the correspondence with the STM experiment is best for low density values, such as 1 × 10^−5^ e/Å^3^ (Figure 3b), as opposed to 1 × 10^−4^ e/Å^3^ or 1 × 10^−2^ e/Å^3^ (Appendix A). This makes sense, since the STM experiment primarily accesses the tail of the wave function. The energy range explored by the STM experiment is also decisive, typically between *E*_F_−0.5 eV and *E*_F_, since a larger interval (involving deeper energy levels) leads to a significantly different picture (Appendix A).

An energy level analysis performed close to the Fermi energy sheds additional light on the properties of the molecular arrangement (see Figure 4a). As expected, the HOMO–LUMO gap of the so-assembled molecules decreases from the single molecule to less than 200 meV, which is in good agreement with the experiment (see Figure 2d). Let us now consider the HOMO and HOMO-1 states relevant for the STM spectroscopy analysis. According to Figure 4b,c, we find that both spin orientations of the HOMO localize on A-type molecules, whereas both spin directions of the HOMO-1 localize on the B-type molecules. The fact that the HOMO wave function is found exclusively on A-molecules, while the HOMO-1 wave function exists exclusively on B-molecules, is an additional confirmation of the electronic (spectroscopic) origin of the checkerboard symmetry observed in the STM contrasts. A wide range density of state (DOS) can be found in Appendix A.

## 4. Conclusions

The very good agreement of the calculation with the experiment tells us that there is no need to consider two sets of conformationally different molecules, as proposed previously. It is found that the molecules are able to arrange themselves so as to exhibit the same alternating contrast observed experimentally and simultaneously, leading to a ϑ = 45° dihedral angle on each molecule of the assembly. Therefore, although we cannot exclude completely a small deviation (less than 2°) of the dihedral angle from 45°, we demonstrate here that the presence of a second species with ϑ = 30° is not a necessary condition to explain the experimental observation. It is noteworthy that good agreement is found between simulation and experiment for a relatively narrow energy window at *E*_F_ and at the low-electron-density calculation, proof that only the tail of the wave function is captured in an STM image. In summary, our approach proves valuable in identifying the STM contrast of LnPc_2_ layers and could be used in similar cases where it is difficult to interpret the STM images of an assembly of molecular complexes.

## Figures and Tables

**Figure 1 nanomaterials-13-02232-f001:**
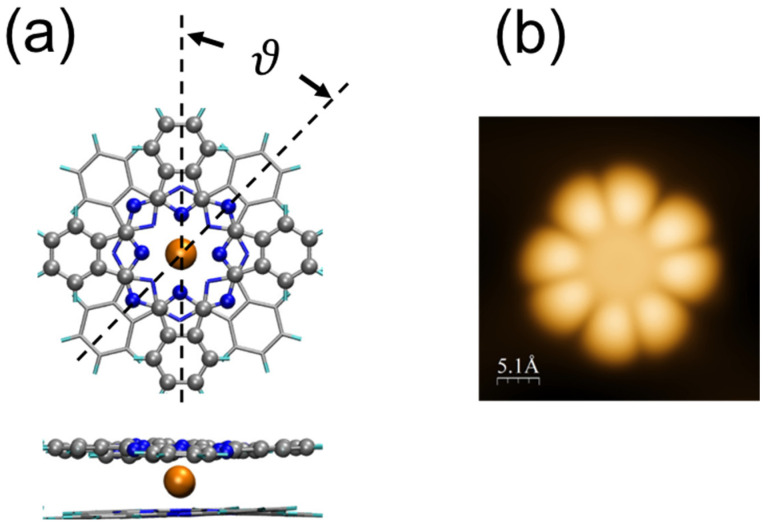
(**a**) Geometry of the TbPc_2_ molecule. ϑ is the dihedral angle between the upper Pc (balls) and the lower Pc (sticks). Color code: C (grey), N (blue), Tb (orange). (**b**) STM image (60 pA, −0.3 V) of a single TbPc_2_ molecule on Au(111), where only the upper Pc is visualized using STM.

**Figure 2 nanomaterials-13-02232-f002:**
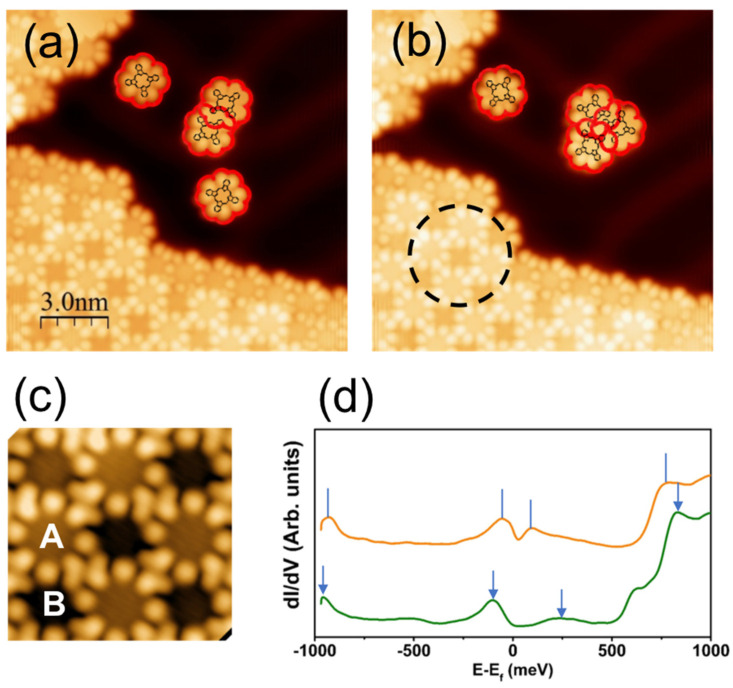
(**a**,**b**) Constant current STM images (70 pA, −0.3 V) of TbPc_2_ molecular islands and clusters grown on Au(111). Contours of upper Pc’s are highlighted in red. (**c**) Zoom-in image of the circled area in (**b**) showing bright (A) and dark (B) contrasts for molecules in a domain (70 pA, −0.3 V). (**d**) dI/dV spectroscopy performed at the center of A-type (orange) and B-type (green) molecules, respectively. Vertical lines/arrows show the correspondence between different peaks. (**a**,**b**) from ref [6] with permission. Further permissions related to the material excerpted should be directed to the ACS.

**Figure 3 nanomaterials-13-02232-f003:**
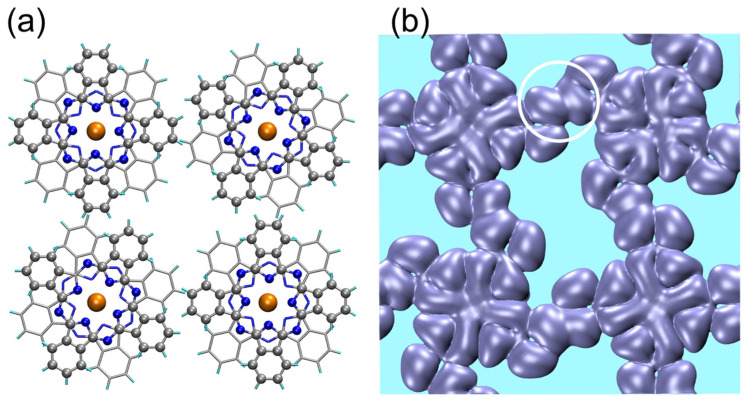
(**a**) Primitive cell 1 × 1 of an assembly of four TbPc_2_ molecules with periodic boundary conditions. Balls: upper Pc; sticks: lower Pc. Color code: C (grey), N (blue), Tb (orange). The only experimental input is the dimension of the square lattice (1.42 nm). After full relaxation, it is found that the dihedral angle ϑ = 45° for all the molecules and remains unchanged independent of initial configuration. (**b**) Electron density 1 × 10^−5^ e/Å^3^ in the range [*E*_F_−0.6 eV, *E*_F_], top view, above the upper Pc. The circle indicates the molecular orbital overlap between adjacent molecules.

**Figure 4 nanomaterials-13-02232-f004:**
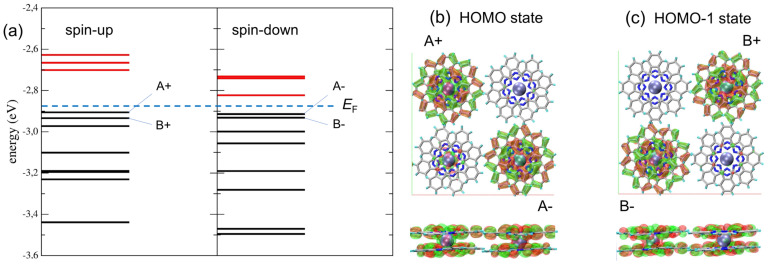
(**a**) Distribution of energy levels close to the Fermi energy (dotted blue line). The levels are labeled A and B to refer to the location of the corresponding wave function with spin up (+) and spin down (−). (**b**,**c**) Top- and side-views of HOMO and HOMO-1 wave functions, respectively. The HOMO-1 state is −0.03 eV below HOMO. Molecule labelling corresponds to the energy levels in (**a**).

## Data Availability

The data presented in this study are available on request from the corresponding author.

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
