# Peer review of "Hierarchical Self-Assembly and Conformation of Tb Double-Decker Molecular Magnets: Experiment and Molecular Dynamics"

_nanomaterials, 2023, doi:10.3390/nano13152232_

Round 1
Reviewer 1 Report
The work reported by Lawes and co-workers provides the way to predict the arrangement of the SMMs on the surface, which is very important to use the SMMs as the molecular magnetic memory. The contents of this work are important, and should be accepted for publication in Nanomaterials.
Author Response
The reviewer finds that our work provides the way to predict the arrangement of the SMMs on the surface, which is very important to use the SMMs as the molecular magnetic memory. He/she considers that the contents of this work are important, and should be accepted for publication in Nanomaterials.
R: We thank the reviewer for a kind evaluation of our work.
Reviewer 2 Report
Minor revisions and editing is needed, keep concepts in similar formatting as much as possible.
The english is good.
Author Response
Reviewer 2: Minor revisions and editing is needed, keep concepts in similar formatting as much as possible.
R: Through careful proofreading of the manuscript, minor issues have been corrected.
Reviewer 3 Report
The manuscript by Laws et al. is a computational investigation of the electronic properties of a monolayer of terbium double-decker single molecule magnet. The adsorption process of magnetic molecules on surfaces is still an unexplored field and there is a lot to discover on how the spin is affected by the interaction with the surfaces. The main idea of the work is interesting and this work could be a first step towards more extensive investigations. However, although all these premises, the work requires some major revisions prior to publication.
In the first place, in the paper, it is claimed that ab initio molecular dynamics have been performed, but it seems that the authors have simply optimized the structure of the monolayer. If this is the case, 'molecular dynamics has to be removed from the title and everywhere it is named in the text. From what it is described, the authors should state that they only performed 'geometry optimizations' and 'geometry relaxations'
The calculations have been performed in the absence of the surface. This is justified by several considerations about the 'inert' nature of the substrate. In some cases, this could be true, but some other reports where the surface plays an active role should be discussed, e.g. Briganti et al. Nano Lett. 2022, 22, 8626-8632 https://pubs.acs.org/doi/full/10.1021/acs.nanolett.2c03161
Since the surface is not explicitly treated, did you apply some constraints on the coordinates? For example on the positions of the terbium atoms.
Have you considered explicitly the 4f electrons of terbium? If so, the 4f DOS could be reported at least in the SI to be compared with other theoretical works.
Why did you choose D2 instead of D3 or rVV10 dispersion correction? At least a single point with D3 should be computed for comparison.
The computed total Density of States of Figure S4 should be compared with the dI/dV spectroscopy of Figure 2d. Both of them could be plotted inside the same graph.
Author Response
We thank the reviewer for the careful reading of our manuscript and the helpful comments and questions. Below, we answer point-by-point all the questions raised by the reviewer and show how we modified the manuscript. Sections that have been added or restated are highlighted in bold characters.
1) In the first place, in the paper, it is claimed that ab initio molecular dynamics have been performed, but it seems that the authors have simply optimized the structure of the monolayer. If this is the case, 'molecular dynamics has to be removed from the title and everywhere it is named in the text.
Damped molecular dynamics has been used to optimize the structure and a complete annealing of the system has been performed before calculating the electronic structure. This is not just a geometric relaxation; therefore, the definition of "molecular dynamics" should be retained and is correct "as is". The text has been slightly amended to make this point perfectly clear.
2) The calculations have been performed in the absence of the surface. This is justified by several considerations about the 'inert' nature of the substrate. In some cases, this could be true, but some other reports where the surface plays an active role should be discussed, e.g. Briganti et al. Nano Lett. 2022, 22, 8626-8632.
As mentioned by the reviewer, the absence of a bearing surface in the simulation is justified by several considerations in the manuscript. However, it is correct that there are experiments where the molecule-substrate interaction cannot be neglected. Examples from the literature are now given including Briganti et al. Nano Lett. 2022, cited by the reviewer. The following sentence has been added:
Line 154: It should be stated however that for other substrates (different systems) the molecule-substrate interaction would not be negligible in the calculations [7,9,10,30]. After full annealing …
3) Since the surface is not explicitly treated, did you apply some constraints on the coordinates? For example, on the positions of the terbium atoms.
No constraint has been applied at all on any atom. As stated in the manuscript, the system is periodically repeated along the planar (x,y) dimensions and this is sufficient to grant stability as observed in the MD annealing procedure.
4) Have you considered explicitly the 4f electrons of terbium? If so, the 4f DOS could be reported at least in the SI to be compared with other theoretical works.
We used a Goedecker-Hutter-Teter semicore pseudopotential including explicitly the following states: 5p6 6s2 4f9 (see quoted reference). This type of pseudopotentials has already been discussed and benchmarked in the literature. Moreover, here we are not interested in the specific projected DOS of the 4f electrons which has anyhow already been reported in former works.
5) Why did you choose D2 instead of D3 or rVV10 dispersion correction? At least a single point with D3 should be computed for comparison.
We are aware of the extension of the Grimme vdW correction D3. However, we have recently shown, see Molecules 2022, 27(24), 9034 (https://doi.org/10.3390/molecules27249034) that the D3 correction is responsible for spurious effects leading even to unphysical phase separation in condensed systems. This strongly discourages its use in modeling extended systems. Furthermore, it is important to be aware of the similarity of the performance of all these vdW corrections apart from the pathological case of D3. On this point, see: J. Phys. Chem. B 2020, 124, 49, 11273–11279. (https://doi.org/10.1021/acs.jpcb.0c08) where rVV10 and several others have been computed and compared, supporting the conclusion above.
To make this point clear, we have added the following sentence in the “computational method” subsection that includes two more references:
Line 98 … we resorted to Grimme’s D2 formula. This vdW correction was preferred to that of D3 which proved to be responsible for parasitic effects leading even to a non-physical phase separation in condensed matter systems [26]. In fact, except for D3, a similarity in performance is found for D2, rVV10 and several others [27].
6) Is it possible to show together DOS and dI/dV?
We did not calculate conductance here, but joint-DOS, which makes comparison dubious.
Round 2
Reviewer 3 Report
I still find that the description of the molecular dynamics part is lacking some specific information. If it has been performed, at least some details should be provided: what is the duration of the trajectory in picoseconds? Which is the timestep? Did you perform thermal annealing? I cannot find these data in the manuscript. If it is a new method, please provide some references.
Author Response
We added the following information to the method section:
The structure was optimized via damped dynamics [30] as implemented in the developer’s version of the CPMD code. The annealing factor of the ions was set to 0.995, and an integration time step of 4.0 (0.097 fs) as well as a fictitious electronic mass of 340 au for the propagation of the electronic wavefunctions within a Car-Parrinello scheme were used, as benchmarked and assessed in all our former publications. A total simulation time of about 8 ps was needed to bring the system to a stress-free structure at 0 K.